# Design and Fabrication of Untethered Light-Actuated Microbots in Fluid for Biomedical Applications

Md Faiyaz Jamil [1] , Mishal Pokharel [2] and Kihan Park [1,*]

1   Department of Mechanical Engineering, University of Massachusetts Dartmouth,
    Dartmouth, MA 02747, USA
2   Department of Biomedical Engineering and Biotechnology, University of Massachusetts Dartmouth,
    Dartmouth, MA 02747, USA
*   Correspondence: kihan.park@umassd.edu

**Abstract:** Untethered mobile robots at the micro-scale have the ability to improve biomedical research by performing specialized tasks inside complex physiological environments. Light-controlled wireless microbots are becoming the center of interest thanks to their accuracy in navigation and potential to carry out operations in a non-invasive manner inside living environments. The pioneering light-engineered microbots are currently in the early stage of animal trials. There is a long way ahead before they can be employed in humans for therapeutic applications such as targeted drug delivery, cancer cell diagnosis, tissue engineering, etc. The design of light-actuated microbots is one of the challenging parts along with the biocompatibility and precision control for in vivo applications. Recent progress in light-activated microbots has revealed a few innovative design concepts. In this study, we presented a framework on the different aspects with a comparative analysis of potential designs for the next generation of light-controlled microbots. Utilizing numerical simulations of fluid-structure interactions, limiting design elements of the microbots are addressed. We envision that this study will eventually facilitate the integration of robotic applications into the real world owing to the described design considerations.

**Keywords:** microbots; microfabrication; optical traps; flow simulations

## 1. Introduction

The pioneering work of Arthur Ashkin led the way toward developing micro and nanostructures for manipulation using light [1]. He developed and instigated the use of optical tweezers to catch and manipulate bacteria and viruses, opening up a new avenue for the light-activation of tiny devices [2]. Since then, researchers have used light to trap and manipulate cells [3,4], synthetic microstructures [5], transfection of genes [6], manipulate organelles within a single cell [7], single cell optoporation [8], spermatozoa isolation [9], protein folding [10] and dynamics [11], and cell membrane characterization [12]. Micro-tools intended for light-mediated biological applications ought to have maneuvering capabilities with an optical trap in real-time and the ability to load and unload cargo using laser-induced thermal convection. The utilization of light for fabrication, active actuation, and control is the defining feature of light robotics. The therapeutic compounds used for targeted therapy can be carried by these microbots in a stable manner, and they can release in exact and controlled amounts at chosen target sites [13–16]. Additionally, by using magnetic fields and light as microbots' actuation sources, wireless microbot mobility and the ability to trigger the release of specific medications are both made possible [17,18]. Light-mediated drug delivery has widely been researched using synthetic and biological components as drug carriers. Ultraviolet, near-infrared light, and up-conversion nanoparticles are primarily investigated for use in theranostics [19].

3D printing emerged in 1986 as stereolithography. Since then, it has been used increasingly in material science engineering to print on metals, glasses, ceramics, polymers,

and tissue engineering substrates [20,21]. The fabrication of micro-substrates has the advantage of high resolution, structure specificity, high aspect ratios along with being time efficient. Light-equipped micro-tools have offered a new approach to carrying biological micro molecules to the intended delivery site [5]. Light actuation of microbots primarily entails a conformational change in the chemical structure of the resin used to create the microbots. Liquid-crystal elastomers in the shape of cylindrical microbots were the first to be propelled in liquid without the use of any external propulsive force or torque. Their movements were propelled by the periodic conformational changes induced by light [22]. These continuum actuators are preliminary self-dependent swimmers and provide a motivational platform for the development of highly adaptable micro-locomotives. A structural alteration in the substrate is caused by the addition of azo-benzene [23] or poly(N-isopropyl acrylamide) (pNIPAM) [24], which then causes the content of the manufactured substrate to be released. Optical actuation of fabricated micro-substrates can provide spatial manipulation, delivery of drugs, and propellant functions of up to 100 μm/s [25,26]. In this study, different types of microcarriers were fabricated for diversity in surface modification and hydrodynamics for testing the property of the mucus barrier. This is required because the active molecule must pass through the intestinal bio-barrier, which is made up of the mucus in the GI tract and a monolayer of intestinal epithelium cells joined by tight junctions in addition to the stomach's acidic environment for oral administration of a medicine to be effective [26]. Indirectly, light-actuated micro-tools can be used to trigger another mechanism and induce phototactic microbot movements such as self-propulsion or self-electrophoresis [27]. In addition, for precision controllability, the microswimmers were loaded with different dyes that allowed the user to control them individually. Prior to this, Tang's team developed an artificial phototactic swimmer that was programmable. The carrier was a Janus-nanotree-structured microswimmer, that contained a nanostructured photocathode and a photoanode at the opposing ends, where cations and anions were discharged, respectively, and was then propelled by self-electrophoresis [28]. Furthermore, light can stimulate photocatalytic [29] or thermoplasmonic [30] responses in microbots. A 3D printed micro dumbbell with two beads and a spike was fabricated and manipulated using holographic optical tweezers, providing possible applications of vibration detection and fluid viscosity measurements [31]. While 3D printing is primarily used for creating intricate structures, drug delivery does not always require complex structures and can rely on the use of microbowls or microspheres for drug-loaded carriers [32]. The collective phototactic behavior of green algae in a solution was imitated by these microswimmers, which self-aligned along the direction of light propagation. Optical modulation of such carriers is far simpler with a focus on target specificity and transport. The fuel-free near-infrared (NIR) propelled gold nano-shells loaded with mesoporous silica nanoparticles demonstrate motion in liquid because of the photothermal effect of the nano-shells [33]. The addition of the gold nano-shells allows the user to control switching on/off the carrier and for remote guidance of it to a specific target for use in diagnostics or drug delivery. Even though substantial research has been done in the enumeration of microbots for use in biological applications, there is a lack of precision systems for drug delivery. In conjunction with the implementation of the proposed designs, fabricated microbots can be used for exactitude delivery of treatment options for diseases in parts of the body that present danger with invasive interference. Researches exploring the use of microtools have their primary focus on either biocompatibility [34], chemical composition [23] of the microtool and physical advantages [35]. There is negligible research on the variation in designs that could affect the use of a microbot in vivo. While similar microtools have been fabricated [25], there is an enormous aperture with regard to mechanically crucial design aspects. It is highly critical that design aspects such as the vehicle's overall weight, feature capability, support functionality, and efficient manipulation by light be considered when fabricating such substrates for biological applications. The proposed designs in this project aim to create a foundation, from which future scholars can relate and move towards the development of microbots that can be used for translational medicine. Furthermore, a state-of-the-art fabrication technique and readily available stable biocompatible materials are used to manufacture the

microbots. The ease of fabrication in conjunction with the design parameters showcases the importance of efficiency and coherence.

This paper aims to illustrate effective designs and methods for 3D printing polymers with micron and nanoscale architectures for use in biomedical engineering applications. It highlights crucial factors to consider when constructing light-powered microbots capable of performing new tasks by discussing the known difficulties and prospective solutions for improved microbot mobility utilizing light. The use of light to control the microbots and the necessary structures required to do so is mentioned with a secondary focus on the addition of features that could enable the propulsion of drug or biological molecules outside the carrier. At the conclusion of the paper, along with the limitations of 3D printing and manipulation using light actuation, the envisioned uses for light-powered microbots are discussed.

## 2. Actuation of Microbots Using Light

Actuation using light exploits the fundamental concept of radiation pressure. The concept of optical trap was developed in the early 1970s [36]. Light is a mode of radiation, and in an optical trap when light passes through a transparent object it gets deflected in different directions. The refracted beams of light cause a momentum balance of the small particles in its way and hold the particles in place in its optical plane. By making the laser beam sharply focused momentum balance in the z-direction can also be attained. To trap the particles of interest, they are preferred to be spherical or symmetric in shape but the efficiency of trapping may depend on the medium where the particles are in [37]. In 1987 A. Ashkin and J.M. Dziedzic showed that they can even capture viruses (Tobacco Mosaic Virus) and bacteria (*E. coli*) using optical traps, which are irregular in shape [2].

The idea of optical tweezers gained attention to manipulate microbots in recent years due to the accessibility to state-of-the-art fabrication techniques that could reach micron or sub-micron levels. This is critical since the applied force in an optical trap is in the pico newton (pN) range. Therefore, it is not possible to lift the large and heavier structures using optical tweezers. Increasing the laser power may compensate for this limitation but for biological use, it can be harmful to the physiological environment depending on the application. Figure 1 illustrates the basic working principle of an optical tweezers system equipped with a spatial light modulator (SLM) for multiple trap generation and manipulation.

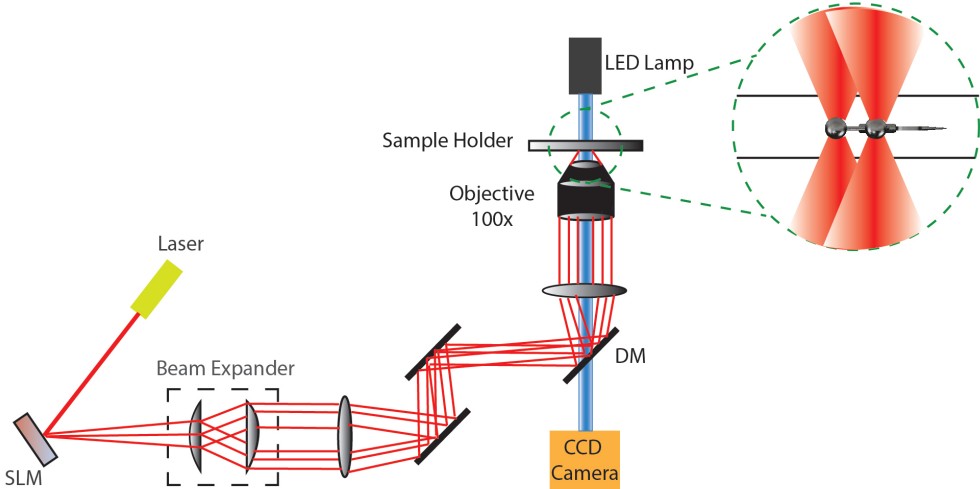

**Figure 1.** Optical trapping principle schematic along with integrated spatial light modulator (SLM) for multiple trap generation and manipulation.

An SLM may be controlled to manipulate the traps by splitting a laser beam into several beams using the computer-generated holography (CGH) technology [38]. Other techniques, such as the use of galvo scanners or acousto-optic deflectors (AODs), are just as effective in creating and managing multiple traps. An AOD mimics several traps and has

the ability to move a single trap quickly using the combined help of acoustic energy with the integrated deflectors. Also, it has the advantage of the laser power not being divided, unlike an SLM, which makes the traps much stiffer than an AOD. But with high-speed reflective phase modulation, the SLM provides a far better resolution to manipulate the optical traps.

## 3. Design of Light-Actuated Microbots

The decision of where to employ a microbot is the first step in an arduous design process. Numerous applications, including biomedical science [39] and micro machinery [40], have made use of the light-actuated microbots. For biomedical applications depending on the field of use, the shape of the body of the microbot has to be considered. There are different predefined curves to select from for minimal drag on the body for flowing conditions. This is important because the optical force that is holding the microbots in place is limited by the physiological constraints and limited laser power for safe in vivo use. A lower center of gravity is also preferred while designing the microbots and ensures better stability and depending on the application could be necessary. For example, if there are some openings at the top of the vehicle then moving in upright order is necessary and a lower center of gravity may help achieve this. Another important aspect of the design is the distribution of the weight symmetrically over the whole body. The distribution of equal weight throughout the body ensures better stability. Symmetric design is desirable as this eliminates the chance to have unbalanced forces acting on the microbot. If the microbot is being used under flowing conditions such as blood flow or induced flow while taking out the fabricated microbots from the substrate or under any other physiological flow situation the structural integrity must be ensured for the carrier to survive. Figure 2 shows a few instances where the microbot fabrication failed due to the lack of strong supports. In this figure, the 3D model is shown followed by the scanning electron microscopy (SEM) images of the fabricated microstructures. An elaborate discussion is presented about the fabrication process in the fabrication of light-actuated microbots section.

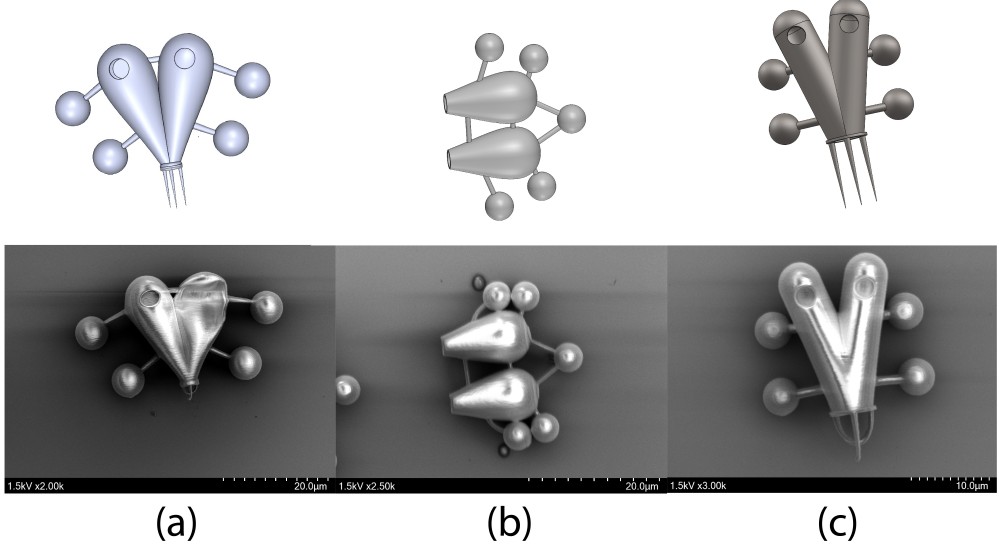

**Figure 2.** SEM images of fabrication limitations of microbots due to the lack of structural integrity (**a**) One of the two channels got deflated during the development process due to lean wall thickness, (**b**) front control handles got deformed due to the flow instigated during the development process, and (**c**) spikes in front of the microbot got deformed due to the lack of support.

### 3.1. Design Constraints

Different conceptual designs can be modeled according to utility, but the fabrication of those designs restricts the actualization. For example, from Figure 2a, the microbot deflated due to the hollow inside which lacks any supports to withstand the outside pressure. Also,

it is not easy to create supports inside to make it stronger as that will limit the functionality of the feature of having additional storage onboard. One solution could be making the thickness of the wall larger, but that can increase the dimension of the microbots and add additional mass to them. Similarly in Figure 2b the front control handler could not withstand the induced flow during the development process. This limitation can be fixed by making the diameter of the connecting rods bigger, but that will again result in additional mass. In optical manipulation we want the mass to be as small as possible for greater control flexibilities. In Figure 2c the front spikes for the poking operation got bent. This occurred because the fabrication is done layer by layer and the front spikes had enough time to sink into the resin during printing, which led to their deformation. These examples should be able to guide through complex design fabrication and actualization of the microbots.

*3.2. Outer Structure*

Designing the body of the microbots is crucial for any application. The design can be started from scratch to meet the desired functionality. But as mentioned before multiple predefined curves can be used for the body of the microbots. To begin with the low drag body design, curves such as a teardrop curve, dumbbell curve, pear curve, piriform curve, etc. can be used before going into the depth of curve optimization process. A recent study by Villanga et al. demonstrated the use of a teardrop-shaped body for light-actuated syringe functionality in a microbot [25]. Following equations can be used in any CAD software to create specific geometrical shapes. Figure 3 represents the different possible curves for the main body design of the microbots. Parametric equation of a teardrop curve [41],

$$x = \cos t \tag{1}$$

$$y = \sin t \sin^m \left( \frac{1}{2} t \right). \tag{2}$$

where, $t$ is the parametric variable of the curve and the enclosed area of the curve is given by,

$$A = 4\sqrt{\pi} \frac{\Gamma\left( \frac{1}{2}(3+m) \right)}{\Gamma\left( 3 + \frac{1}{2}m \right)}. \tag{3}$$

here, $A$ is the enclosed area and $m$ is an input parameter.

Another interesting curve is the sextic dumbbell curve. A dumbbell curve can be expressed using the following implicit equation [42],

$$a^4 y^2 = x^4 \left( a^2 - x^2 \right) \tag{4}$$

here, $a$ is an input parameter and the size of the curve depends on it, but not the shape. The enclosed area of this curve is,

$$A = \frac{1}{4} \pi a^2 \tag{5}$$

Another choice of the body design could be derived from the pear curve [43]. For a specific set of $r$ values in the following equation, the Mandelbrot set lemniscate $L_3$ in the iteration towards the Mandelbrot set is a pear curve. The following implicit equation can project the pear curve for a constant value of $r$,

$$r^2 = (x^2 + y^2)(1 + 2x + 5x^2 + 6x^3 + 6x^4 + 4x^5 + x^6$$
$$- 3y^2 - 2xy^2 + 8x^2y^2 + 8x^3y^2 + 3x^4y^2 + 2y^4 + 4xy^4 + 3x^2y^4 + y^6) \tag{6}$$

Another distinct curve form the pear curve is a pear-shaped curve. The curve is given by the following Cartesian equation [44],

$$b^2y^2 = x^3(a - x). \tag{7}$$

And it has the area of,

$$A = \frac{a^3\pi}{8b} \tag{8}$$

A similar curve close to the pear-shaped curve is the piriform curve. It is a quartic algebraic curve also known as the peg-top curve [45],

$$a^4y^2 = b^2x^3(2a - x) \tag{9}$$

And the area is,

$$A = \pi ab \tag{10}$$

which is exactly the same as the ellipse with semiaxes *a* and *b*.

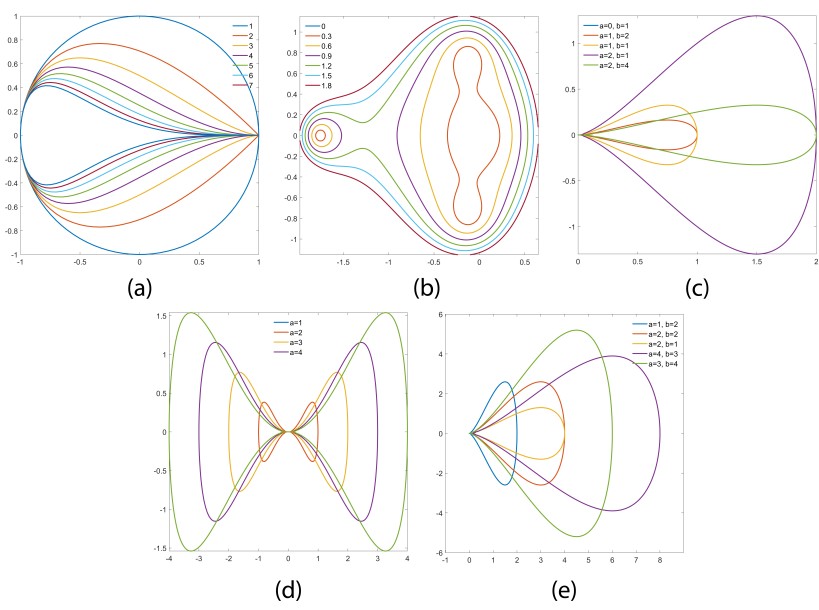

**Figure 3.** Different curves for the microbot's body design (**a**) teardrop curve ($m = 1 - 7$), (**b**) pear curve (*r* varied from o to 1.8 with 0.3 interval), (**c**) pear-shaped curve (with varying *a* and *b* values), (**d**) dumbbell curve ($a = 1 - 4$), and (**e**) piriform curve (with varying *a* and *b* values).

### 3.3. Control Handlers

Design considerations for control are vital for proper manipulation of the microbots. Although an optical trap can trap objects of irregular shapes, it can trap spherical objects most efficiently [37,46]. An optically controlled microbot needs to have some kind of handles for precision control and spherical control handles are preferable over any other irregular shapes because of the aforementioned reason. In this section, the best possible configurations are discussed for better control. To avoid obstacles in the way of carrying out the microbots' primary objective, it is preferable to spread the control handles out to lessen optical trap interference between them. Also, how many control handles should there be in the design is the following concern. This depends on the application and maneuver requirements. If the microbot only stays in one place then one spherical handle is enough, but this can cause rotation in a flowing environment. Adding two spherical handles ensures better stability to hold it in place. Adding more handles is better for stability but this also increases the overall weight of the microbot and occupies a larger area that can also increase the drag in the flowing condition. For this reason, many studies suggested using three to four handles as an optimal number of the control handles for light-actuated microbots [30,47,48].

### 3.4. Additive Features Onboard

With the body-making process being complete, the next is adding selective and desired features to it according to the application. Different features depending on the use of the microbot can be added, for example, to achieve local heating or to create a flow locally for mixing fluids a heating disk in the body of the microbot may be useful. A heating disk is basically a metal layer on top which creates heat and consequently convection current when the laser is shined upon the disk. Along with the natural convection, the Marangoni effect may be dominant to create a strong convection current [49,50]. Marangoni effect appears when there is a difference between surface tensions and creating a local temperature gradient by using the laser and heating plate a strong current can be generated. Local micro level heating may be necessary to agitate a biological environment or to induce movement for better mixing by increasing the Brownian motion. A microbot can be equipped with a spike at the front for measuring the viscoelastic properties of a cell. Literature suggests that the elasticity measurement using microbots is much more accurate for softer cells than the conventional method of using Atomic Force Microscopy (AFM) [48]. By comparing the elasticity measurements to the relevant available data of a healthy cell, it is possible to determine whether a cell is malignant or if there is any irregularity in the cell structure. Additionally, micro tools like this have a tremendous deal of potential for doing investigations on cell-to-cell interaction with extreme accuracy and selectivity. Fabricating complex scaffolds for delivering multiple entities is crucial when it comes to regenerative medicine, tissue engineering, therapeutics, and theranostics. 3D printing makes the fabrication of such scaffolds easier with faster speeds and higher specificity. When repairing the articular cartilage, scaffolds used are larger and are seeded with cells, growth factors, hyaluronic acid, hydroxyapatite [51]. Microbots equipped with multiple channels can be used to load growth factors which can be delivered into the implanted scaffold site for quickening the healing process. The size of the multi-channel carrier can also be upscaled or downscaled as per requirement. Stimulating a cell by delivering messenger ribonucleic acid (mRNA), makes it produce a specific protein. mRNA's can be delivered using microbots to target cells for inducing the production of a specific protein that would help therapeutically [52]. Multiple channels can be added to a microbot so that it can carry more payloads or do multiple tasks at once. In this event even if one of the channels is compromised (Ref. Figure 2a), this safeguards the delivery of the cargo. Also, different applications may require to have multiple channels such as delivering two different types of cargo which are reactive to each other. The only thing that has to be ensured in this case is that, whether the radiation pressure of light is strong enough to carry the cargo while complying with the physiological environment.

## 4. Fabrication of Light-Actuated Microbots

### 4.1. Nanoscale 3D Printing

Several 3D printing methods can be utilized to fabricate microbots. One such technology uses two-photon polymerization (TPP) to precisely fabricate nano and microstructures for biological applications. The structures shown in Figure 4 were made using Nanoscribe Photonic Professional jul (Germany) at the nano lab at Massachusetts Institute of Technology, Cambridge, MA, USA. It is possible to create filigree structures with practically any 3D shape using this equipment, including crystal lattices, porous scaffolds, organically inspired patterns, smooth curves, sharp edges, undercuts, and bridges [53]. The GT2 allows the user to control the sub-micron feature size in all spatial directions. The structures were designed using Solidworks and were converted into a job file by the Describe software used by the GT2 printer. The software allows the users to use slicing and hatching functions for defining the resolution and precision of printing. The Describe software's job file specifically ordered the 3D printer stage to move in the three directions (x, y, and z) with the desired locations and speeds.

### 4.2. Materials

A wide variety of resins are available for use in the Nanoscribe GT2 Professional 3D printer such as IP-PDMS, IP-n162, IP-Visio, IP-Q, IP-S, IP-Dip, IP-G, and IP-L. All the mentioned resins are provided by Nanoscribe. For printing microbots with nanostructures, it was essential that the resolution was at its highest therefore we chose the $63\times$ objective which gives the highest possible printing resolution. IP-Dip resin was chosen for this application, primarily because of its biocompatibility, non-cytotoxicity, and for printing sub-micron features with high aspect ratios [54].

### 4.3. Fabrication Process

A pinch of IP-Dip resin was poured onto a 10 mm $\times$ 10 mm silicon chip and placed inside the printer. A $63\times$ objective was used to focus the laser onto the silicon chip and print the structures. The printing of the microbots could be visualized with an in-built X-ray vision camera. After completion of the print, the microbots were put in a beaker containing propylene glycol monomethyl ether acetate (PGMEA) for 20 min for developing the print. Thereafter, the silicon chip containing the microbots was carefully placed in isopropyl alcohol (IPA) bath for 5 min to get rid of any excess PGMEA. The microbots were viewed under a microscope for confirmation of print before taking SEM images.

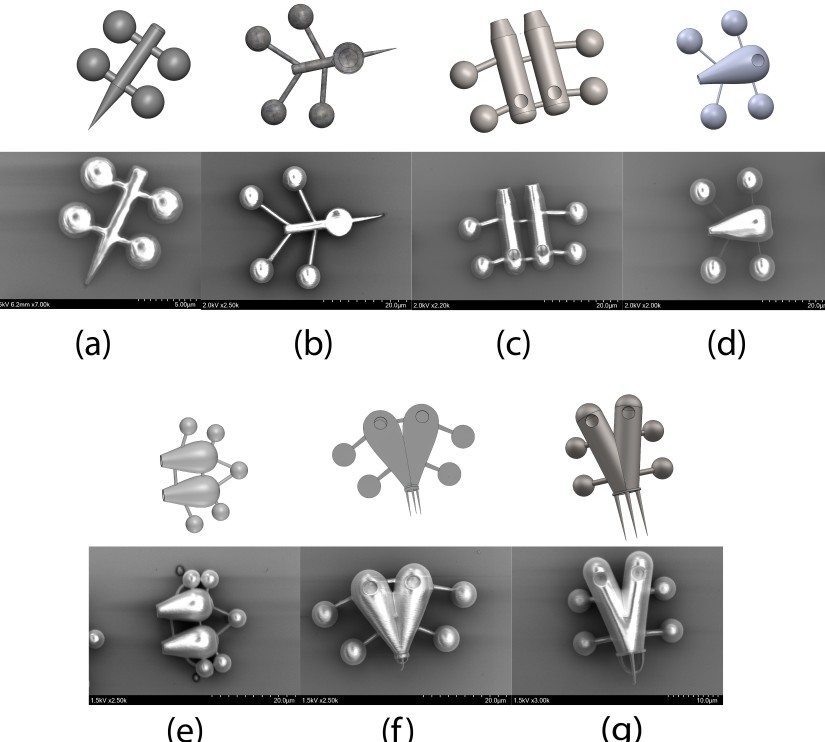

**Figure 4.** Fabricated microbots along with the 3D models followed by the SEM images (**a**) a simple microbot with four control handles and a spike at the front to poke cells or any object of interest, (**b**) a microbot equipped with a heating disk and a front poking mechanism, (**c**) microbot with a double channel with double opening design (straight), (**d**) microbot with single channel design, (**e**) double channel with double opening design (teardrop curve), (**f**) double channel with single opening design and spikes at the front (teardrop curve), and (**g**) double channel with single opening design and spikes at the front (straight).

## 5. Simulation for Fluid-Structure Interaction (FSI)

In this section, we conducted flow simulations in ANSYS and Solidworks to visualize the fluid flow and fluid-structure interaction (FSI) inside the hollow body of the microbots. FSI simulation results help in visualizing the weak spots of a structure when exposed to a flowing condition. In this study, the sole purpose of the FSI computation and fluid flow

simulations is to give an insight to the limitations of the fabrication and failure mechanism of the structures. It shows where the maximum pressure develops and consequently suggests reinforcing those areas for better fabrication of the microbots without disrupting any functionality. For FSI analysis we used ANSYS fluid flow (CFX) and structural analysis. In CFX, to simulate a possible realistic flowing condition, we used shear stress transport (SST) with a total energy transfer model. The fluid domain temperature was set to 310 K which is close to a human body temperature and water was chosen as the working fluid for this simulation since it closely resembles blood flowing conditions. The inlet velocity is kept constant at 1 m/s$^2$ which is close to the maximum velocity of blood in major arteries and the outlet condition was set to a constant pressure of 16 KPa which is the maximum systolic blood pressure in a healthy human body [55,56]. The dimensions of the fluid domain were chosen in such a way that any dimension of the fluid domain is 10× the maximum dimension of the microbot in that direction. This eliminates the interference of the boundary effect to the calculated results. The wall condition is kept smooth and adiabatic for the microbot.

Figure 5 shows the pressure distribution over the body of the microbots. From this figure, it can be seen that wherever the pressure builds up, a similar failure is observed during the fabrication process. Therefore, these structural components must be sturdy enough to withstand the incoming flow.

From the FSI data, it indicates that with the aforementioned flowing conditions, pressure develops in the front extended arms for the single channel design with a maximum value of 20 KPa which is 4 KPa higher than the reference environment pressure of 16 KPa. In a similar fashion, pressure builds up in the front region of the double channel with a single opening. From Figure 5b,c, the affected regions (marked with a red circle) due to the pressure build-up from flowing fluids can be seen. The tips of the spikes of the model got deformed during the development process and strong supports are required at the bottom to hold them in place. In this simulation, the maximum observed pressure is found to be 16.57 KPa. In the double channel dual opening (Figure 5d,e) the slender connecting rods to the spheres got bent due to the induced flow during the development process. This can be eliminated either by making a thicker connecting rod or by creating supports at the far side of the spheres to give additional strength to the structure. The maximum pressure, in this case, is computed at around 16.55 KPa.

Also, we used Solidworks to simulate the flow due to the rise in temperature in a hollow channel which effectively acts as a loading mechanism for the microbots. In this simulation, we tried to show how the convection current can be used to load cargo inside the body of the microbot. We kept the surrounding fluid medium stagnant and only one heating source was added to one of the two channels to demonstrate a comparison of the effect. From Figure 6, it is clearly visible that the convection current starts as soon as the heating source is activated. The contour plot here indicated the velocity of the flowing fluid. In a realistic scenario during the fabrication of microbots, this heating element can be a layer of metals that can generate heat with laser irradiation [30]. A comprehensive fabrication technique to create this layer is described in a previous study [25].

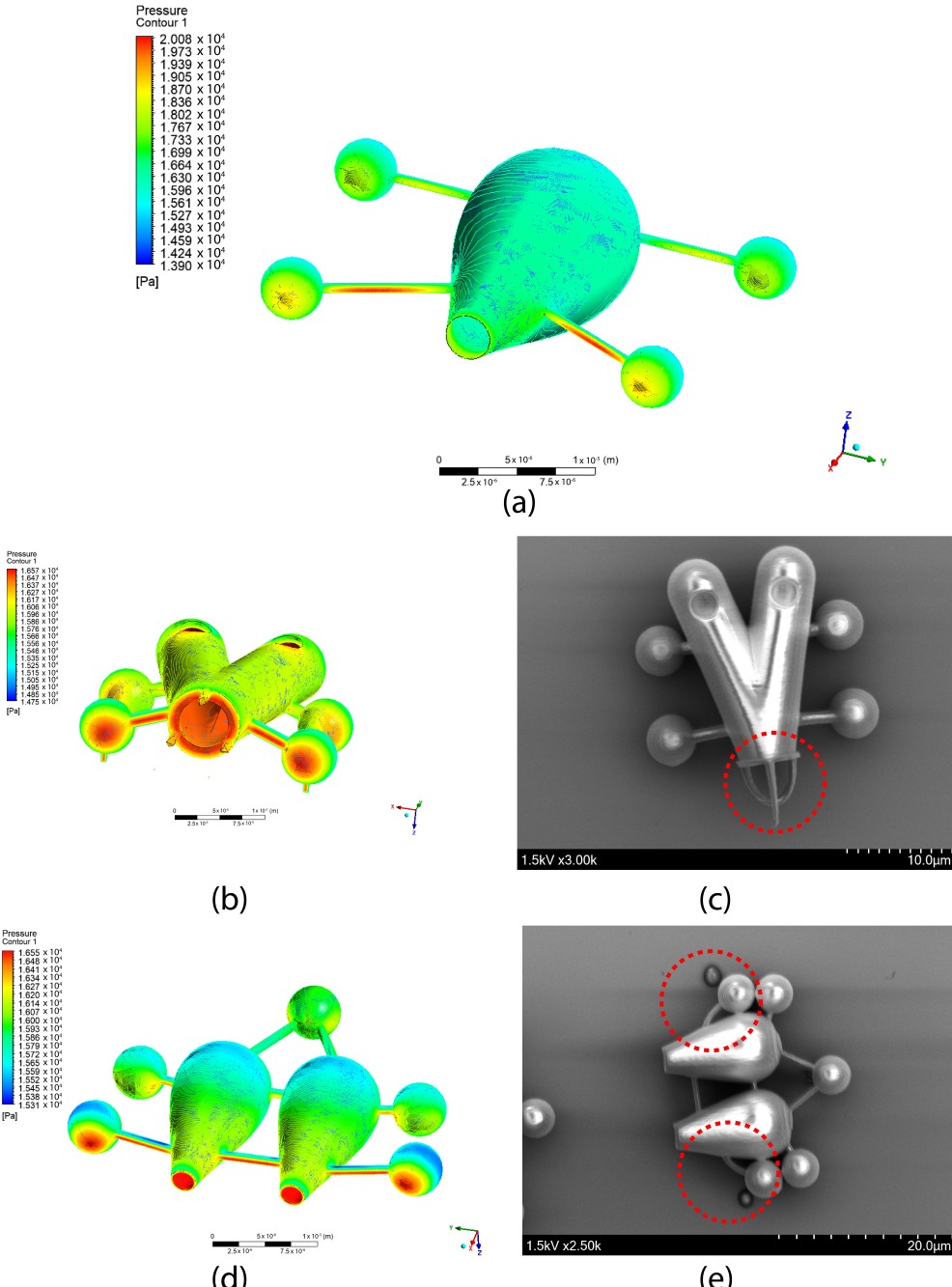

**Figure 5.** Computational simulation results of fluid-structure interactions (FSI) with the corresponding limitation of fabrication for (**a**) single channel design, (**b**) double channel design with spikes and single opening in front, (**c**) failure (red circle) in the delicate features (spikes) at the front during the development process due to lack of support, (**d**) double channel design with a dual opening at the front, and (**e**) corresponding failure (red circle) of the design due to pressure build-up during the development process.

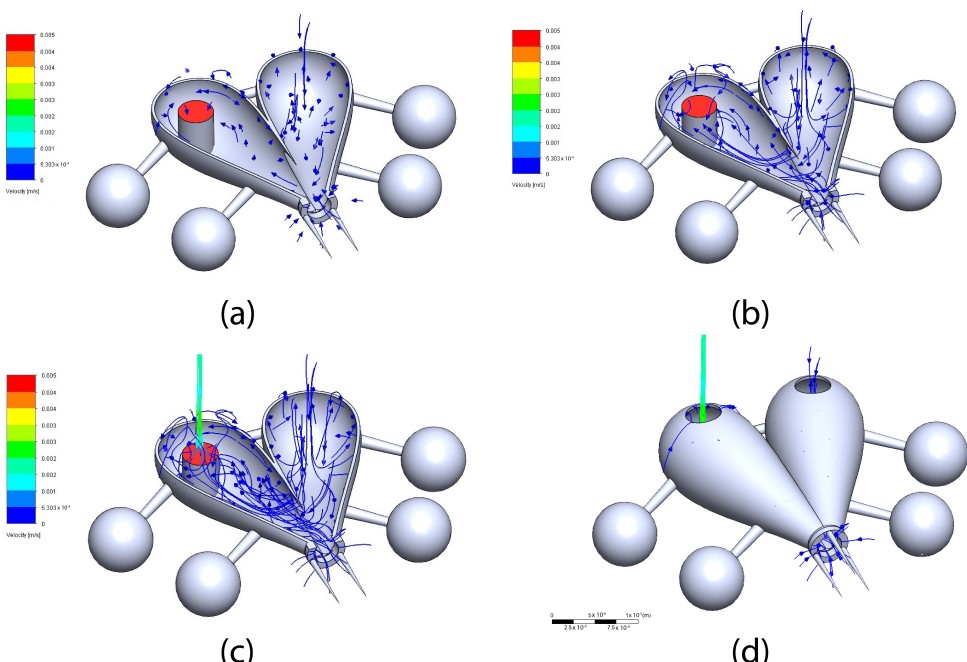

**Figure 6.** Simulation results of fluid flow due to the convection current created by a heating source (red marked) inside the microbots (sectional view) at different time stamps (**a**) 3 s, (**b**) 6 s, (**c**) 10 s, and (**d**) microbot model without the cross-section.

In this simulation, the heating element was kept as a 100 mW constant heating source and the walls were kept at adiabatic condition. The computational domain was kept 10× the maximum dimension of the microbot in any direction to eliminate any boundary effects. The maximum dimension of the microbot model is 40 microns and the maximum fluid velocity was found 2–3 mm/s for ideal conditions.

## 6. Conclusions

Light manipulation at the sub-micron level is gaining more focus with the improved fabrication techniques and the vision to use them in biomedical applications. It is anticipated that collaboration across various research disciplines connected to robotics at the micro-scale will grow in the near future, leading to novel ideas and fixes to present difficulties due to the growing interest in robotics at the micro-scale that has become apparent in the past few years. To do that, a model to direct the microbots is required, but there aren't any of them to the best of our knowledge. In this study, we elaborately discussed 3D modeling and the state-of-the-art fabrication techniques of light-actuated microbots with different design considerations. Designing can be critical at the sub-micron or nano level, and for biomedical applications, additional biocompatibility has to be met to avoid any contamination. We also presented simulation studies with fluid-structure interactions and flow simulations (conducted in ANSYS CFX and Solidworks) considering the fluid flow in the physiological environments where these microbots may be employed in the future. The simulation results can explain why there were manufacturing flaws with the microbots and what measures may be taken to reduce them. In a future study, we will present the collection and exportation of the microbots from the substrate to the workspace and their manipulation under the optical tweezers assembly. Furthermore, we plan to incorporate drugs into the microbots and release them using NIR light. We hope that this article will serve as a reference for robotics researchers in the future who are interested in the advancement of light-activated microbots in biomedical applications. Light-actuated microbots will be crucial in cargo delivery to cellular environments with bio-inspired designs and innovative materials. It is vital to focus on microbots being cost-effective and environment-friendly as well.

**Author Contributions:** Conceptualization, M.F.J., M.P. and K.P.; methodology, M.F.J. and M.P.; software, M.F.J.; data analysis, M.F.J. and M.P.; investigation, M.F.J. and M.P.; resources, K.P.; writing—original draft preparation, M.F.J. and M.P.; writing—review and editing, M.F.J., M.P. and K.P.; supervision, K.P.; project administration, K.P.; funding acquisition, K.P. All authors have read and agreed to the published version of the manuscript.

**Funding:** This research work is supported by UMass Dartmouth's Marine and Undersea Technology (MUST) Research Program funded by the Office of Naval Research (ONR) under Grant No. N00014-20-1-2170.

**Institutional Review Board Statement:** Not applicable.

**Informed Consent Statement:** Not applicable.

**Data Availability Statement:** The authors declare that any data related to this study will be provided upon request.

**Acknowledgments:** The authors acknowledge the MIT.nano facility for the fabrication upon the institutional agreement with the University of Massachusetts Dartmouth and Dapeng Li for the support with the SEM machine at the University of Massachusetts Dartmouth.

**Conflicts of Interest:** The authors declare no conflict of interest.

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
