# Peer review of "Design and Fabrication of Untethered Light-Actuated Microbots in Fluid for Biomedical Applications"

_2673-3161, doi:10.3390/applmech3040071_

Round 1
Reviewer 1 Report (Previous Reviewer 2)
The paper is well structured and suitable for publication in Applied Mechanics. A few minor questions need to be addressed.
(1) In the abstract keyword "Vivo applications" is used VIVO has several meanings so the author should not use confusing words.
(2) Concerning English authors need to proofread paper from a native speaker.
Reviewer 2 Report (Previous Reviewer 3)
`The authors have incorporated well the previously told minor problems in the current version.
This manuscript is a resubmission of an earlier submission. The following is a list of the peer review reports and author responses from that submission.
Round 1
Reviewer 1 Report
The current manuscript discusses the design and fabrication of untethered light actuated microbots in fluid for biomedical applications. From the overall read of the manuscript please find below suggests / comments to enhance the paper further before it can be considered for publication.
1. In the literature review section (section 1), it will be good to identify research gaps within this field and how this study is trying to fill some of these gaps
2. It will be good to identify design constraints along with functional requirements for these microbots. Based on these, the authors can generate several conceptual designs and have them tested in a simulation. Optimize the design variables through the simulations to obtain the desired functional / operational requirements. Based on this optimized design, the authors should fabricate them (3D) printing and discuss the quality of the print and how the printing parameters can be optimized to achieved a desired print. Such reporting will add significant interest and knowledge to the field of study
3. For the finite element analysis, how was the model validated?
Reviewer 2 Report
The manuscript is suitable for publication in Applied Mechanics. I recognize that the topic is important, and the content, style, and format are suited for “Applied Mechanics”. The manuscript entitled “Design and fabrication of untethered light-actuated microbots in fluid for biomedical applications” is an interesting topic and will take part to contribute to research in biomedical applications.
Reviewer 3 Report
The paper is well structured. Few minor questions need to be addressed.
1- In the abstract key words, Optical Traps is used, however no explanation or content is described in the text. Explain
2- What different aspects and limiting design elements are referred throughout the paper. Clear highlights should be mentioned.
3- What lack of strong supports the authors are highlighting that allowed micro fabrication failed in section 3.
4- Add few more relevant references in each section.
5- Explain in detail the FSI results and what is achieved from the results in computational fluid dynamic domain.